# Polyamines in Ovarian Aging and Disease

**DOI:** 10.3390/ijms242015330

**Published:** 2023-10-18

**Authors:** Bo Kang, Xin Wang, Xiaoguang An, Chengweng Ji, Weikang Ling, Yuxin Qi, Shuo Li, Dongmei Jiang

**Affiliations:** 1State Key Laboratory of Swine and Poultry Breeding Industry, College of Animal Science and Technology, Sichuan Agricultural University, Chengdu 611130, China; xinwang@stu.sicau.edu.cn (X.W.); anxiaoguang@stu.sicau.edu.cn (X.A.); jichengweng@stu.sicau.edu.cn (C.J.); lingweikang@stu.sicau.edu.cn (W.L.); qiyuxin@stu.sicau.edu.cn (Y.Q.); 2023202044@stu.sicau.edu.cn (S.L.); 2Farm Animal Genetic Resources Exploration and Innovation Key Laboratory of Sichuan Province, Sichuan Agricultural University, Chengdu 611130, China

**Keywords:** polyamine, ovary, aging, autophagy, disease, cancer

## Abstract

Ovarian aging and disease-related decline in fertility are challenging medical and economic issues with an increasing prevalence. Polyamines are a class of polycationic alkylamines widely distributed in mammals. They are small molecules essential for cell growth and development. Polyamines alleviate ovarian aging through various biological processes, including reproductive hormone synthesis, cell metabolism, programmed cell death, etc. However, an abnormal increase in polyamine levels can lead to ovarian damage and promote the development of ovarian disease. Therefore, polyamines have long been considered potential therapeutic targets for aging and disease, but their regulatory roles in the ovary deserve further investigation. This review discusses the mechanisms by which polyamines ameliorate human ovarian aging and disease through different biological processes, such as autophagy and oxidative stress, to develop safe and effective polyamine targeted therapy strategies for ovarian aging and the diseases.

## 1. Introduction

Polyamines, including putrescine, spermidine, and spermine, are a class of polycationic alkylamines widely distributed in mammals at millimolar concentrations [1]. Putrescine, spermidine, and spermine become positively charged at physiological pH levels. The interaction between polyamines and negatively charged biological macromolecules directly or indirectly affects the structural stability and spatial conformation of DNA, RNA, phospholipids, and proteins, and regulates different biological processes such as DNA transcription, RNA methylation, ion channel regulation, and post-translational modification of proteins [2,3]. Polyamines can also induce autophagy, remove ultra-physiological doses of reactive oxygen species (ROS), and inhibit oxidative stress, apoptosis, and inflammation [4,5]; however, the underlying mechanisms haves not been fully uncovered. An increasing number of studies have shown that dietary polyamines can prevent diseases and extend lifespan by activating autophagy, improving mitochondrial metabolism, and regulating maintaining polyamine pools in tissues [6,7].

Over the past few decades, the issues associated with ovarian aging and disease have progressively escalated in nearly all developed nations. Individuals increasingly opt to delay childbirth, and women’s ovaries age earlier than other organs. Most women suffer a significant decline in reproductive capacity after 37 years of age, and pregnancy rarely occurs after 45 years of age, causing various medical and social issues [8]. In 2023, scientists published a paper summarizing the factors contributing to ovarian aging and suggested solutions to delay reproductive aging in females and preserve fertility [9]. It is important to explore these solutions and improve female fertility. Polyamine metabolism has long been considered a potential therapeutic target for aging and disease. Polyamines are directly related to aging. Although regulating intracellular polyamine pool homeostasis has been extensively studied, little is known about the relationship between polyamines and ovarian aging and disease. In addition, the biological function of polyamines, with anti-inflammatory, antioxidative stress, and autophagy-promoting properties, can delay ovarian aging and diseases by regulating polyamine metabolism [10]. Therefore, it is necessary to understand polyamine homeostasis in human ovarian aging and disease to develop a safe and effective polyamine-targeted therapy strategy.

## 2. Polyamine Metabolism

Polyamines are primarily obtained through food intake, with a minor portion originating from intestinal microorganisms and endogenous metabolic pathways [11,12]. Mammalian cells precisely regulate polyamine pool homeostasis through polyamine biosynthesis, transmembrane substance transport, and polyamine catabolism [13,14]. S-adenosylmethionine (SAM) is converted to decarboxylated S-adenosylmethionine (dcSAM) by S-adenosylmethionine decarboxylase, which provides a substrate for the synthesis of spermidine and spermine [15,16]. Arginine is converted to ornithine by arginase to provide a precursor. Ornithine decarboxylase (ODC) decarboxylates ornithine to produce putrescine. Then, aminopropyl is added by spermidine synthase (SPDS) and spermine synthase (SPMS) to produce spermidine and spermine [17,18]. As a homodimer of ODC, ornithine decarboxylase antizyme (OAZ) can regulate polyamine levels through post-transcriptional regulation. OAZ produces ODC-OAZ heterodimer, whereas 26S protein bodies recognize inactive ODC-OAZ heterodimer and degrade it in a ubiquitin-independent manner [19,20]. The decomposition pathway of polyamines can be characterized by acetylpolyamine oxidase (APAO) and spermine oxidase (SMOX), respectively. APAO requires spermidine/spermine N1-acetyltransferase (SSAT) synergy [21]. First, SSAT processes polyamines to form N^1^-acetylspermine and N^1^-acetylspermidine, which can be further metabolized by APAO (Figure 1). In addition, the SMOX family directly oxidizes spermine to produce spermidine and other metabolites such as 3-aminopropionaldehyde and H_2_O_2_. Polyamines can inhibit the expression of SSAT mRNA transcript, which contains an exon with a premature stop codon. Thus, high expression of SSAT promotes the acetylation of spermidine and spermine, converting excess spermidine or spermine into putrescine or transporting it to the extracellular space.

Spermidine participates in the hypusination of eukaryotic translation initiation factor 5A (eIF5A). It is the only known cellular protein containing the essential amino acid hypusine (Nε- (4-amino-2-hydroxybutyl) lysine). It mainly acts on a sequence encoding a specific peptide motif that promotes the translation extension of mRNA and assists in translation termination by stimulating the hydrolysis of peptide tRNA [22]. First, deoxyhypusine synthase (DHPS) transfers the 4-aminobutyl of spermidine to a specific lysine residue ε-amino to generate an intermediary molecule (Lys50 in humans). Second, deoxyhypusine hydroxylase (DOHH) immediately and irreversibly catalyzes the hydroxylation reaction, and deoxyhypusine is converted to active hypusine [23]. This is essential for regulating the translation and post-translational modification of autophagy-related proteins [24,25] (Figure 2).

## 3. Polyamines Improve Ovarian Aging

Ovarian aging can be divided into two categories: physiologic ovarian aging and pathologic ovarian aging, both characterized by menstrual disorders and decreased fertility [26]. Ovarian aging is a complex process that may depend on various factors. Oxidative stress, apoptosis, inflammation, mitochondrial dysfunction, and telomere depletion can influence ovarian aging [27]. Recently, the antiaging effects of polyamines have been demonstrated in age-related cardiovascular diseases [28], neurodegenerative diseases [29], musculoskeletal diseases [30], and immune diseases [31]. However, similar effects may be obtained in ovaries. Here, we will discuss the effects of polyamines on the major markers of ovarian aging and explore their potential to prevent ovarian aging and alleviate age-related ovarian diseases.

### 3.1. Polyamines Mitigate Oxidative Stress and Apoptosis in Ovaries

Oxidative stress and apoptosis are closely related to ovarian aging [32,33]. Apoptosis of oocytes and female germline stem cells directly reduces the number of germ cells, decreases ovarian reserve, and reduces the reproductive potential [34]. In addition, massive apoptosis of granulosa cells disrupts metabolic homeostasis in the ovaries and impairs ovarian function [35]. Reactive oxygen radicals are excessively produced in aged ovaries, leading to structural damage and apoptosis. Therefore, antioxidants may possibly inhibit ovarian oxidative stress and prevent ovarian aging [36]. Jiang et al. [37] fed ICR mice with spermidine-containing water for 3 months and found that the number of atretic follicles and tissue levels of malondialdehyde significantly decreased and the activities of antioxidant enzymes increased. These findings suggest that spermidine can increase cellular antioxidant capacity and improve ovarian function. There are relatively few studies on how polyamines alleviate oxidative stress. There has been evidence that feeding spermidine to female mice can activate the NRF2 pathway of ovarian granulosa cells to promote the synthesis of antioxidant enzymes and inhibit 3-nitropropionic acid (3-NPA) induced oxidative stress [38]. Meanwhile, Jiang et al. [39] found that the use of chloroquine (autophagy inhibitor) could eliminate the antioxidant effect of spermidine, indicating that spermidine could relieve oxidative stress through autophagy. Studies have found that spermine can act as a free radical scavenger to inhibit H_2_O_2_ production and protect DNA from oxidative damage [40,41]. At the same time, through transcriptional sequencing, we found that KITLG, CD44, and RND2 associated with oxidative stress were significantly altered by exogenous spermidine supplementation in the oxidative stress model established by 3-NPA (unpublished manuscript). At the same time, overexpression of ODC significantly increased granulosa cell activity, elevated the Bcl-2/Bax ratio, and inhibited apoptosis [42]. Based on existing studies, it can be implied that polyamines can exert antiaging functions through antioxidant and antiapoptotic mechanisms. For example, Xu et al. [43] found that supplementation with spermidine and spermine can prevent apoptosis in the aging brain by increasing Bcl-2 expression and superoxide dismutase (SOD) activity, decreasing the protein expression of Bax and Caspase-3, and reducing the production of malondialdehyde. These effects are similar to those of the antiaging drug rapamycin. Thus, polyamines, with antioxidant and antiapoptotic effects, may play an antiaging role in ovarian aging.

### 3.2. Polyamine Induces Autophagy in Ovaries

Impaired autophagy is the hallmark of aging [44]. In humans and animals, impaired nonselective autophagy exacerbates tissue damage, whereas spermidine ameliorates the impaired state of cellular autophagy and effectively prolongs cell viability [45,46]. Previously, it has been found that an increased activity of granulosa cells is associated with enhanced levels of autophagy in ovaries [47]. Jiang et al. [37,39] demonstrated that drinking spermidine-containing water upregulated LC3 II and Beclin 1 expression in the ovaries of ICR mice and induced autophagy by upregulating LC3 II/LC3 I ratio and downregulating p62 expression in granulosa cells and ovaries. We further revealed that spermidine-induced autophagy can alleviate 3-NPA-induced oxidative stress and autophagy to improve ovarian function. Oral administration of spermidine upregulating LC3 II expression and autophagic flux in the murine heart significantly prolonged the lifespan of mice and exerted cardioprotective effects, whereas cardioprotective properties of spermidine was lost in mice lacking ATG5 in their cardiomyocytes [48]. These findings suggest that the antiaging effect of spermidine is mediated through autophagy.

eIF5A is a key factor for autophagosome formation [49,50]. Inhibition of hypusinated eIF5A (hyp-eIF5A) with GC7 significantly decreased the expression of TFEB, a factor controlling the biogenesis of autophagosomes and lysosomes, and inhibited autophagic vesicle formation [51]. Hyp-eIF5A levels decrease with aging, but dietary spermidine can increase hyp-eIF5A levels and activate autophagy [52,53]. Han et al. [54] found that chrysin inhibited eIF5A expression and autophagy in protozoan cells, but exogenous spermidine restored the expression of genes related to autophagy and hyp-eIF5A. Interference with SPMS in B lymphocytes significantly decreased the intracellular concentrations of spermine and hindered autophagy. In addition, treatment with exogenous spermine enhanced autophagy, whereas eIF5A knockdown impeded autophagy. ODC gene knockout or SPMS inhibition by difluoromethylornithine (DFMO) decreased TFEB expression, which was reversed by exogenous spermine. Autophagy core factor ATG3 is essential for the maturation of autophagosomes, and the mutation of key motifs in ATG3 markedly impairs its translational regulation by eIF5A [50]. It has been revealed that eIF5A plays an important role in regulating cellular autophagy. The mRNA level of eIF5A was significantly upregulated after hCG-induced downregulation of LHR, whereas eIF5A level decreased over time, suggesting that eIF5A is involved in the post-transcriptional regulation of ovarian LHR mRNA-binding protein [55]. Therefore, we hypothesized that spermidine mediates hyp-eIF5A modification and regulates ovarian autophagy to inhibit ovarian aging, but in-depth studies are needed to unravel the specific mechanisms.

### 3.3. Polyamines Inhibit Inflammation in Ovaries

Long-term chronic inflammation accelerates aging [56]. Accumulating evidence suggests that chronic inflammation is strongly associated with aging [57]. A study by Timóteo-Ferreira et al. [58] demonstrated that higher levels of inflammatory markers such as TNF-α and IL-1β exist in the ovaries of aged mice. Thus, inflammation may similarly play an important role in ovarian aging. Spermidine plays a crucial role in decelerating aging, as it alleviates inflammation. Pretreatment with spermidine suppressed the LPS-induced release of nitric oxide, prostaglandin E2, TNF-α, and IL-1β, and suppressed the LPS-induced nuclear translocation of NF-κB in murine RAW264.7 macrophages [59]. Oral administration of spermidine and spermine reduced the expression of NLRP3, IL-18, and IL-1β and alleviated inflammation in the brain of SAMP8 mice, decelerating brain aging and Alzheimer’s disease progression [43]. Therefore, spermidine may reduce the expression of inflammatory factors to inhibit aging-induced inflammation of ovaries.

### 3.4. Polyamines Alleviate Telomere Damage in Ovaries

Telomeres are responsible for maintaining genomic integrity and chromosomal stability [60]. Wirth et al. [61] assessed relative telomere length in cardiac tissue sections from young mice, aged mice, and aged spermidine-treated mice by quantitative fluorescence in situ hybridization (Q-FISH). Compared with aged mice, spermidine-treated mice had a significantly reduced proportion of short telomeres, while the number of Q-FISH spots with high fluorescence intensity was significantly reduced in the nuclei of aged mice, indicating that the length of some telomeres in aged mice was less than the detection threshold in the experimental conditions. Meanwhile, the telomere length of aged mice, but not aged spermidine-treated mice, was shorter than that of young mice. However, the mechanism by which spermidine preserves telomere length is still unclear. Previous studies have shown that the activation of the Nrf2 signaling pathway or the oxidative stress inhibitor NAC can increase telomerase expression and activity, preserve the number of primordial follicles, and improve the quality of oocytes in old mice [62,63]. Interestingly, spermidine can activate the Nrf2 signaling pathway, increase the activity of antioxidant enzymes and alleviate oxidative stress [37,64]. Therefore, spermidine may preserve telomere length and prevent ovarian aging by attenuating oxidative stress.

### 3.5. Polyamines Improve Mitochondrial Function

Granulosa cells, as the largest cell population in the ovary, require abundant and stable mitochondria to provide sufficient energy for their growth, proliferation, and division [65]. However, intracellular mtDNA damage significantly increases with granulosa cell proliferation, evidenced by reduced mitochondrial membrane potential and mitochondrial dysfunction. Mitochondrial dysfunction ultimately reduces granulocyte activity and cell cycle arrest [66,67]. Mitochondrial dysfunction is one of the main causes of aging. A recent study found that spermidine and spermine concentrations significantly decrease in aging hearts, and spermidine supplementation can increase the total polyamine pool of the heart and significantly prevent cardiac aging [68]. Exogenous spermidine inhibited mitochondrial ROS production and mitochondrial damage [68]. In addition, spermidine supplementation improved mitochondrial mass in vascular endothelial cells and promoted angiogenesis in senescent mice [69].

SIRT1, PGC-1α, and NRF1 promote mitochondrial biogenesis, and sustained mitochondrial biogenesis contributes to longevity [70,71]. The expression of SIRT1, PGC-1α, and NRF1 decreases with aging, which is positively correlated with ODC and negatively correlated with SSAT. Exogenous spermidine upregulated the SIRT1/PGC-1α pathway in aged mice, suggesting that spermidine can alleviate aging by enhancing mitochondrial biogenesis through the SIRT1/PGC-1α pathway [68,72]. In addition, proteome sequencing showed that spermine protected mitochondrial function, upregulated mitochondrial proteins, and enhanced mitochondrial respiration in the brains of aged drosophila [53]. Meanwhile, spermidine promoted hyp-eIF5A in the brain. Inhibition of hyp-eIF5A by GC7 impaired mitochondrial function and accelerated mitochondrial damage [53]. Thus, polyamines not only alleviate age-related mitochondrial damage, but also promote mitochondrial biogenesis to delay ovarian aging.

### 3.6. Polyamines Improve Oocyte Quality in Ovaries

It has been observed that putrescine can affect oocyte quality, and that putrescine deficiency is a major cause of poor egg quality in aged mice [73]. Putrescine supplementation reduced egg aneuploidy, improved the quality of embryos, and reduced the abortion rate in aged mice [74,75]. ODC activity was significantly lower in the ovaries of aged mice than in young mice, and putrescine-containing drinking water significantly reduced oocyte aneuploidy [73]. Meanwhile, putrescine supplementation during oocyte maturation improved the quality of oocytes in senescent mice [76]. Shi et al. [77] used putrescine to treat the ovaries of senescent mice, and found that it improved the quality of oocytes and reduced the incidence of aberrant spindle and chromosomal aneuploidy. These studies demonstrate that putrescine supplementation in aged mice can effectively improve oocyte damage caused by ovarian senescence.

In summary, polyamines can suppress aging-associated oxidative stress, apoptosis, inflammation, and telomere shortening and maintain mitochondrial biogenesis and function partly through autophagy. Although more studies are needed to unfold the effect of polyamines on ovarian senescence, previous findings suggest that polyamines might protect against ovarian aging (Figure 3).

## 4. Polyamines and Ovarian Disease

Ovarian diseases, including ovarian cancer, polycystic ovary syndrome (PCOS), primary ovarian insufficiency (POI), and luteal phase deficiency (LPD), can threaten women’s health and cause infertility [78]. Currently, there are few studies on polyamines in other ovarian diseases except for ovarian cancer. In the following section, we mainly review the relationship between ovarian cancer, a common ovarian disease, and polyamines and unfold the potential association of polyamines with other ovarian diseases.

### 4.1. Polyamines and Ovarian Cancer

Ovarian cancer, the second most common gynecologic cancer, is one of the deadliest cancers among women [79]. Polyamine levels are significantly higher in cancer cells compared to normal cells [80]. Thus, the polyamine metabolic pathway is a promising target for treating cancer [81]. ODC is the rate-limiting enzyme for polyamine synthesis and a direct transcriptional target of MYC. The majority of cancers exhibit a significant increase in MYC expression, which in turn promotes ODC expression to expedite polyamine biosynthesis [10,82,83]. Because of the direct link between polyamines and oncogenes, polyamine metabolism plays a crucial role in cancer treatment [84]. SSAT modulates the homeostasis of polyamines and can be used as a cancer biomarker [85]. Using liquid chromatography–mass spectrometry, Niemi et al. found a significant increase in polyamine levels in the urine of patients with ovarian cancer. Thus, urinary polyamine levels can be used as a disease biomarker [86]. Below, we have summarized the treatments for ovarian cancer.

#### 4.1.1. DFMO-Targeted Inhibition of ODC for the Treatment of Ovarian Cancer

DFMO covalently binds to the ODC activation site to inhibit polyamine biosynthesis and prevent tumor development [10]. Hwang et al. [87]. found that ODC activity gradually decreases as DFMO concentration increases. DFMO gradually decreased the activity and proliferation ability of SKOV-3 cells and induced the expression of proapoptotic proteins, Bax, and cleaved Caspase-3 to promote the apoptosis of SKOV-3 cells. The DFMO-mediated inhibition of cell proliferation was alleviated by exogenous polyamines. Cisplatin and DFMO co-treatment synergistically promoted SKOV-3 cell death [87]. Poly ADP ribose polymerase (PARP) repairs damaged DNA, as polyamines promote the enzymatic activity of PARP and facilitate its binding to DNA [88]. Using the immunofluorescence assay method, El Naggar et al. found that after hydrogen peroxide-induced DNA damage in ovarian cancer cells, 1 mM DFMO strongly inhibited PARylation and enhanced DNA damage in ovarian cancer cells to reduce cell viability. Simultaneous exposure of ovarian cancer cells to DFMO and rucaparib (PARP inhibitor) inhibited polyamine synthesis and PARP activity, thereby enhancing the killing effect of cisplatin on ovarian cancer cells. These findings suggest that DFMO can be used as an adjuvant chemotherapeutic agent in the treatment of ovarian cancer [89].

#### 4.1.2. Spermine Analogs Treat Cancer by Reducing Cellular Polyamine Concentrations

Spermine analogs are widely used in cancer treatment [90]. Spermine analogs mainly compete with natural polyamines for uptake. They accumulate in the intracellular space to promote polyamine catabolism and reduce polyamine biosynthesis [91]. Spermine symmetrically receives N-terminal ethyl groups to form the spermine analog N^1^, N^11^ -bis(ethyl)norpermine (BENSPM) [92]. BENSPM, also named DENSPM, is a polyamine analog that efficiently induces SSAT expression and stabilizes SSAT activity. Cotreatment of human ovarian cancer cell line A2780 with platinum-based drugs and BENSPM increased the translation and stability of SSAT, leading to polyamine depletion and decreasing cellular activity [93]. SPB-101, a spermine analog that is structurally similar to BENSPM, can affect the enzymatic activity of SSAT and ODC to promote polyamine catabolism and inhibit polyamine synthesis. It affects the activity of ovarian cancer cells and restricts tumor progression [94,95]. The intracellular concentration of polyamines is tightly controlled, and spermine analogs are highly promising in treating ovarian cancer.

#### 4.1.3. Targeted Modulation of SSAT Enzyme for Treating Ovarian Cancer

An acute induction of SSAT expression induces tumor cell growth arrest and apoptosis [96]. SSAT levels are normally low in cells, but the gene expression, protein level, and enzyme activity of SSAT significantly increase during tumorigenesis [97]. Marverti et al. found that berberine can promote the expression of SSAT in the human ovarian cancer 2008 cell line and C13* cell line, increase the enzymatic activity of SSAT to promote the degradation of polyamines in cancer cells, and inhibit cancer cell growth [98]. In addition, treatment of both cell lines with quinoline increased the production of reactive oxygen species, promoted the expression of SSAT, and decreased polyamine levels, leading to cancer cell death [99]. Overexpression of SSAT inhibited the growth of human ovarian carcinoma cell line C13*, led to spermine depletion, and increased sensitivity to spermine analog N^1^, N^12^-bis(ethyl)spermine [100]. Tummala et al. [93] showed that the combination of two platinum drugs and BENSPM can induce SSAT gene expression and increase its enzymatic activity in A2780 human ovarian cancer cells, leading to spermine and spermidine depletion in A2780 cells and inducing cancer cell death. In addition, the combination of platinum-based drugs and BENSPM improved BENSPM absorption by cancer cells and improved their anticancer effect [93]. However, the combination of platinum-based drugs and BENSPM cannot always improve efficacy due to the decreased uptake of platinum by platinum-resistant cells.

#### 4.1.4. Other Treatments for Ovarian Cancer

In addition to the above treatments, other drugs can also treat ovarian cancer. SI-4650 can inhibit the activity of spermine oxidase to inhibit the development of ovarian cancer [101]. Spermine-coupled lipophilic Pt(IV) precursors can treat ovarian cancer by binding to mitochondria [102]. Therefore, targeted therapy for spermine oxidase and mitochondrial function may be a potential research direction for treating ovarian cancer.

Various drugs can target the metabolism of polyamines and reduce their production to inhibit tumorigenesis. Studies have shown that the deletion of autophagy-related genes, such as Beclin 1 and Atg7, contributes to tumorigenesis in specific tissues and organs [103]. Coni et al. found that hyp-eIF5A promotes MYC expression by alleviating the stagnation of MYC mRNA on the ribosome, thereby promoting colon cancer cell growth [104]. In addition, they found that simultaneous inhibition of ODC and eIF5A can downregulate MYC expression and decrease colon cancer cell growth [105]. Gobert et al. demonstrated that spermidine supplementation can increase eIF5A hypusination in intestinal epithelial cells to prevent the development of colon cancer [106]. Spermidine can promote autophagy and target eIF5A hypusination to regulate the expression of autophagy-related proteins [49,51]. Therefore, SPD-mediated hypusination of eIF5A can be used in future studies to regulate the expression of autophagy-related proteins and oncogenes, such as MYC, JUN, and FOS, in ovarian cancer (Figure 4).

### 4.2. Polyamines and Other Ovarian Diseases

The biological functions of polyamines, such as their anti-inflammatory, antioxidant, and antiaging properties, can indirectly modulate other ovarian diseases. We previously found that polyamines regulate the secretion of estrogen and progesterone, thereby affecting the reproductive ability of animals [107]. In addition, polyamines improve the antioxidant capacity of ovaries and reduce ovarian oxidative damage in animals [37]. Therefore, polyamine supplementation is a potential treatment strategy for ovarian diseases.

#### 4.2.1. Polyamines May Be a Potential Treatment for PCOS

PCOS is a common and complex endocrine disease in women of childbearing age, with an incidence rate of 5% to 10% [108,109]. The clinical symptoms of PCOS mainly include polycystic ovaries, ovulation disorder, hyperandrogenism, and obesity [110]. Oxidative stress is a pathogenic mechanism of PCOS. ROS levels were measured by detecting fluorescence intensity in leukocytes obtained from patients with PCOS. A significantly elevated ROS level was observed in the leukocytes of PCOS patients compared to the control group [111]. Glutathione and oxidized glutathione levels were measured using high-performance liquid chromatography in patients with PCOS. There was a significant reduction in blood glutathione levels and the glutathione/oxidized glutathione ratio in patients with PCOS compared with control subjects [112]. Polyamines decreased ROS levels and protected against oxidative stress. Additionally, they promoted the expression of antioxidants, such as GSH and SOD, thereby augmenting the antioxidant capacity [113]. Therefore, polyamines may eliminate ROS and induce GSH expression in patients with PCOS to alleviate oxidative damage and enhance the antioxidant capacity.

#### 4.2.2. Potential Roles of Spermidine in Primary Ovarian Insufficiency

POI, also known as premature ovarian failure, refers to the loss or dysfunction of ovarian follicles characterized by amenorrhea before 40 years of age [114]. The main clinical symptoms include sleep disorders, depression, reduced fertility, and cardiovascular disease. Studies have shown that mitochondrial dysfunction may induce POI, and improving mitochondrial function may prevent POI. Mutations in individual genes and chromosomal abnormalities are important causes of POI. Tiosano et al. found that mutated genes associated with POI, such as MRPS22, POLG, TWNK, and LARS2, are all involved in mitochondrial DNA replication, gene expression, and protein synthesis and degradation [115]. Notably, spermidine plays an important role in improving mitochondrial function. Eisenberg et al. have found that spermidine can improve mitophagy in murine cardiomyocytes and protect the heart [48]. In addition, the researchers found that feeding spermidine significantly improved hippocampal mitochondrial function and enhanced mitochondrial respiration [116]. Overall, spermidine has a potential therapeutic effect on improving mitochondrial function and cardiovascular disease, and this suggests that spermidine may play a regulatory role in the occurrence of POI.

#### 4.2.3. Polyamines Are Involved in Luteal Phase Deficiency

LPD is the result of the insufficient production or storage of progesterone by the luteum, and its clinical symptoms are shortened menstrual cycles or infertility [117]. LPD is a clinical manifestation of luteal defects caused by various physiological and pathological mechanisms, including abnormal follicle development, hormone secretion disorders, etc. Previous studies have shown that LPD is closely related to the secretion of estrogen and progesterone in the luteal phase, leading to abnormal follicle development [118]. The distribution of ovarian ODC and spermidine during the rat estrous period was consistent with that of steroidogenic acute regulatory protein. In addition, treatment with putrescine significantly increased the serum levels of E2 during rat proestrus [119]. Pregnant horse serum gonadotropin (PMSG) and human chorionic gonadotropin (hCG) significantly increased murine serum E2 and P4 levels, but there was no significant increase in serum E2 and P4 levels in mice concomitantly receiving PMSG, hCG, and DMFO. These findings suggest that DFMO inhibits ODC activity and blocks the positive effects of PMSG and hCG on murine ovarian E2 and P4 [120]. Our studies have found that polyamines can promote the secretion of estrogen and progesterone by regulating steroidogenesis and improving the reproductive ability of animals [107]. Polyamine-mediated ovarian steroidogenesis may help treat LPD. Therefore, supplementing exogenous polyamines may be a potential therapeutic strategy for LPD.

## 5. Outlook

In this work, combined with the results of our research team’s studies on the biological functions of polyamines, we reviewed recent research advances on polyamines in ovarian aging and disease. The review consisted of two main parts: the alleviation of ovarian aging by polyamines and resistance to ovarian diseases. At the same time, we also reviewed the metabolic pathways of polyamines in eukaryotic cells in recent years. Furthermore, we also made a prospect on the potential regulatory role of polyamines in animal ovarian function.

Although polyamine-induced autophagy has been shown to improve heart, brain, and gut health, there are few studies on polyamine-induced autophagy in ovarian aging and diseases. Previous studies by our group showed that spermidine can alleviate oxidative stress and apoptosis in ovarian granulosa cells by inducing autophagy. In our study, spermidine ameliorated ovarian fibrosis and restored mitochondrial morphology and function in aged mice. Therefore, we speculate that polyamine can induce autophagy to delay ovarian aging and treat ovarian diseases. But there are still several areas that require further research in the future: (1) polyamines are involved in the regulation of ovarian aging through various pathways, which need to be further investigated; (2) most studies investigated the antiaging effects of polyamines in vivo and in vitro; therefore, future clinical trials are needed to explore the safety and efficacy of polyamines in aging; (3) antiporters have been shown to improve oocyte quality; thus, whether spermidine and spermine ameliorate ovarian aging needs to determined. Finally, the significant scientific achievements of recent years and the many new discoveries and mechanisms still require careful attention and additional studies.

## Figures and Tables

**Figure 1 ijms-24-15330-f001:**
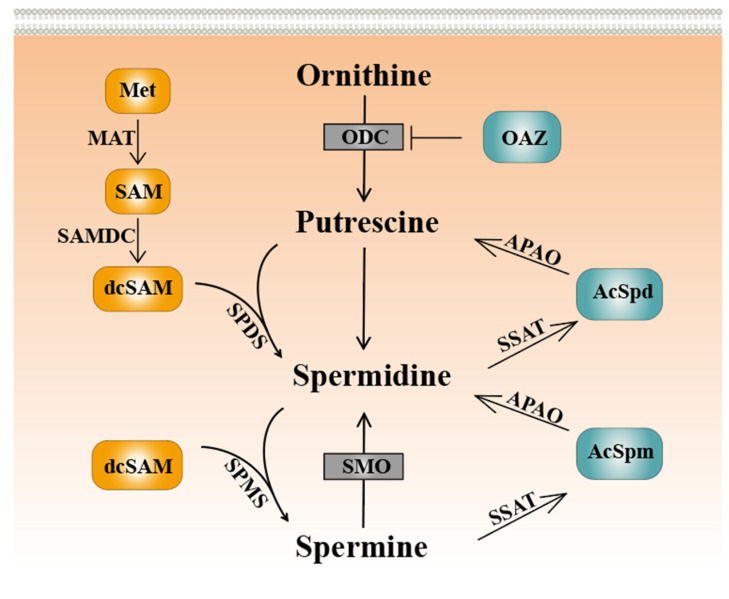
Schematic figure of the core polyamine metabolic pathway in eukaryotes. Met: methionine. MAT: S-adenosylmethionine synthase. SAM: S-adenosylmethionine. SAMDC: S-adenosylmethionine decarboxylase. dcSAM: decarboxylated S-adenosylmethionine. SPDS: spermidine synthase. SPMS: spermine synthase. SMO: spermine oxidase. OAZ: ornithine decarboxylase antizyme. ODC: ornithine decarboxylase. APAO: acetylpolyamine oxidase. AcSpd: acetylspermidine. SSAT: spermidine/spermine N1-acetyltransferase. AcSpm: acetylspermine.

**Figure 2 ijms-24-15330-f002:**
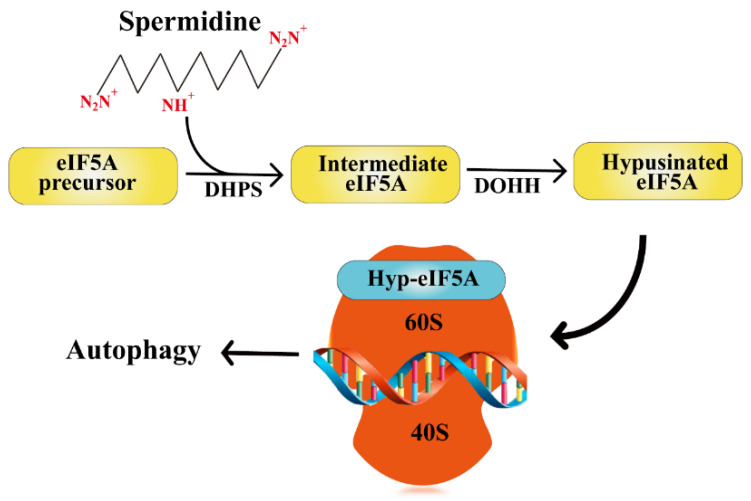
Spermidine mediates eIF5A hypusination to promote autophagy. DHPS: deoxyhypusine synthase. DOHH: deoxyhypusine hydroxylase.

**Figure 3 ijms-24-15330-f003:**
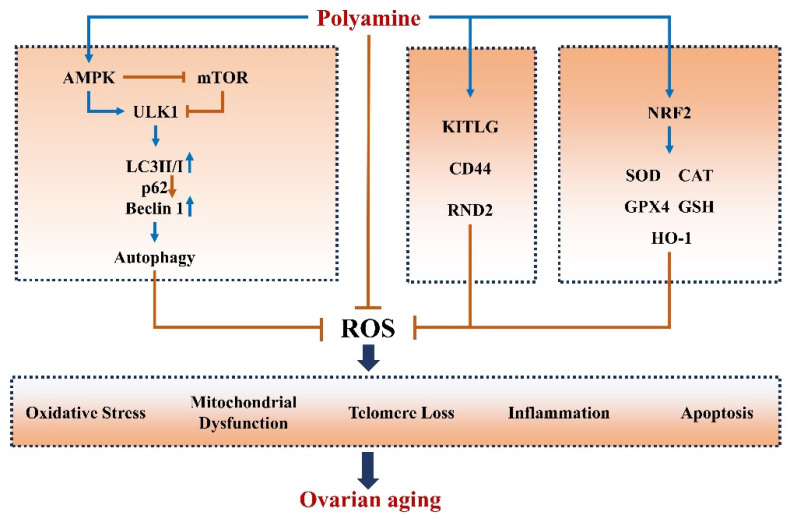
Polyamines prevent ovarian aging. Ovarian aging is associated with oxidative stress, apoptosis, inflammation, mitochondrial dysfunction, telomere loss, and autophagy, which are functionally correlated with each other. Spermidine may alleviate ovarian aging through various mechanisms, such as promoting autophagy, attenuating oxidative stress and inflammation, enhancing mitochondrial biogenesis, and preventing telomere shortening. AMPK: adenosine 5′-monophosphate (AMP)-activated protein kinase. mTOR: mammalian target of rapamycin. ULK1: human autophagy initiation protein 1. KITKG: KIT ligand. CAT: catalase. GPX4: recombinant glutathione peroxidase 4. GSH: L-glutathione. HO-1: heme oxygenase.

**Figure 4 ijms-24-15330-f004:**
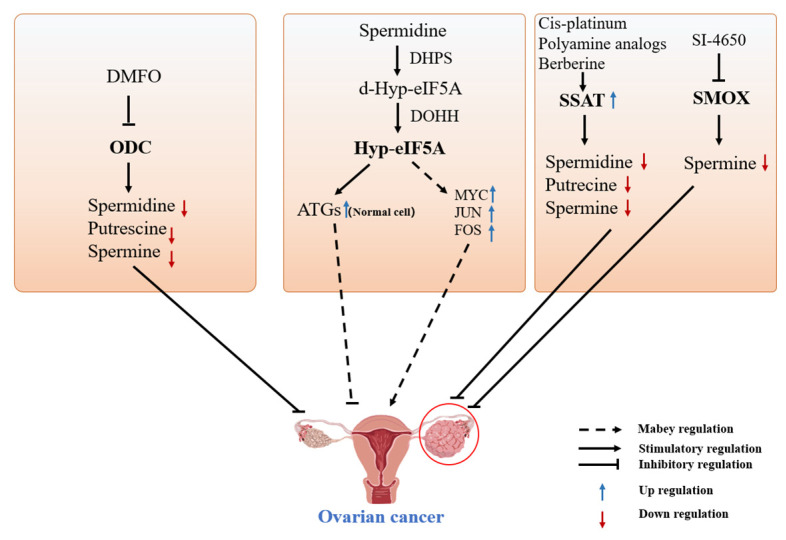
Polyamine metabolism regulates the development of ovarian cancer. (Left) Inhibition of ODC, a key target of anabolic enzymes in polyamine metabolism, can hinder the development of ovarian cancer. (Middle) In normal cells, spermidine may regulate the translation of some autophagy-related genes by mediating hyp-eIF5A, thereby inhibiting the development of ovarian cancer. Spermidine may promote the development of ovarian cancer by regulating oncogene translation through hyp-eIF5A. (Right) Anticancer drugs inhibit the development of ovarian cancer by targeting the catabolic enzymes of polyamine metabolism, such as SSAT and SMO.

## Data Availability

All data generated or analyzed in this study are included in this paper and can be obtained from the authors upon reasonable requests.

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
