# Peer review of "Polyamines in Ovarian Aging and Disease"

_ijms, 2023, doi:10.3390/ijms242015330_

Round 1
Reviewer 1 Report
The manuscript is a well written concise review presenting the mechanisms of effects of polyamines in ovarian aging, cancer, and some other diseases, including modulation of oxidative stress, apoptosis and autophagy, inhibition of inflammation, alleviation of telomere damage and improvement of mitochondrial function. The application of spermine analogs for ovarian cancer treatment is also discussed. The review is based on 119 literature references. Polyamine metabolism is also briefly presented.
Remarks:
The antioxidant action of polyamines could be discussed in more detail. The authors mention augmentation of activities of antioxidant enzymes by polyamines but more mechanisms have been proposed (although polyamines are not directly acting antioxidants).
Line 32: “haves”, should be “have”
Line 363: GSH is not a protein
Generally fine. I noticed one obvious error (line 32) but it might be a typographical error.
Author Response
Dear Reviewer-1,
Thank you for your comments concerning our manuscript entitled “Polyamines in ovarian aging and disease” (ID: ijms-2661765). Those comments are all valuable and very helpful for revising and improving our paper, as well as the important guiding significance to our researches. We have studied comments carefully and have made correction which we hope meet with approval. The main corrections in the paper and the responds to your comments are as following:
Yours sincerely,
Bo Kang
E-mail: bokang@sicau.edu.cn
Response to Reviewer-1,
Comments 1: The antioxidant action of polyamines could be discussed in more detail. The authors mention augmentation of activities of antioxidant enzymes by polyamines but more mechanisms have been proposed (although polyamines are not directly acting antioxidants).
Response 1: Thank you for pointing this out. I agree with this comment. Therefore, the antioxidant effects of polyamines have been described and discussed in more detail in this paper. According to the literature we reviewed, most of the current studies only involve the enhancement of antioxidant enzyme activity by polyamines, with relatively few studies on their pathways of action. We added the following description from line 121 to 132.
Comments 2: Line 32: “haves”, should be “have”
Response 2: Agree. I have revised accordingly the word (Line 32).
Comments 3: Line 363: GSH is not a protein
Response 3: Agree. I have revised accordingly the sentence (Line 380).
References
- Niu, C.; Jiang, D.; Guo, Y.; Wang, Z.; Sun, Q.; Wang, X.; Ling, W.; An, X.; Ji, C.; Li, S.; et al. Spermidine suppresses oxidative stress and ferroptosis by Nrf2/HO-1/GPX4 and Akt/FHC/ACSL4 pathway to alleviate ovarian damage. Life sciences 2023, 332, 122109, doi:10.1016/j.lfs.2023.122109.
- Jiang, D.; Wang, X.; Zhou, X.; Wang, Z.; Li, S.; Sun, Q.; Jiang, Y.; Ji, C.; Ling, W.; An, X.; et al. Spermidine alleviating oxidative stress and apoptosis by inducing autophagy of granulosa cells in Sichuan white geese. Poult Sci 2023, 102, 102879, doi:10.1016/j.psj.2023.102879.
- Ha, H.C.; Sirisoma, N.S.; Kuppusamy, P.; Zweier, J.L.; Woster, P.M.; Casero, R.A., Jr. The natural polyamine spermine functions directly as a free radical scavenger. Proceedings of the National Academy of Sciences of the United States of America 1998, 95, 11140-11145, doi:10.1073/pnas.95.19.11140.
- Sava, I.G.; Battaglia, V.; Rossi, C.A.; Salvi, M.; Toninello, A. Free radical scavenging action of the natural polyamine spermine in rat liver mitochondria. Free radical biology & medicine 2006, 41, 1272-1281, doi:10.1016/j.freeradbiomed.2006.07.008.
Reviewer 2 Report
The review is well described, based on the authors` and others experience. It consists of two main parts: the alleviation of ovarian aging by polyamines and the resistance to ovarian diseases.
I don`t have any negative comments.
This review discusses the mechanisms by which polyamines ameliorate human ovarian aging and disease through different biological processes, such as autophagy and oxidative stress, to develop safe and effective polyamine targeted therapy strategies for ovarian aging and the diseases. The topic of manuscript is proper and address a specific gap in the field. So far, there was no a unified description of the action of polyamines in ovarian aging and ovarian disorders. This review provides such description. The conclusion is clear and well-defined. The references cited by authors in this review are appropriate. There are 4 figures which explain polyamines` action - it helps the reader follow the text.Author Response
Dear Reviewer-2,
Thank you for your comments concerning our manuscript entitled “Polyamines in ovarian aging and disease” (ID: ijms-2661765). Those comments are all valuable and very helpful for revising and improving our paper, as well as lead the way to our researches.
Yours sincerely,
Bo Kang
Reviewer 3 Report
Dear the Editor
Kang B et al reviewed the physiological roles of polyamines during regenerative process. Role of reactive oxygen species was briefly discussed in aging process. In contrast, the role of polyamines was mainly discussed on ovarian cancer based on gene regulation rather than oxidative stress. Role of polyamine on prostaglandin E2 seemed obscure.
Major concerns:
1) In Fig. 3, it seemed unclear the role of other antioxidants. It is important to described how polyamines can act in text, if this is important. Is Fig 3 essential for this manuscript?
Minor concerns:
1) There are many typos and grammatical errors (ie L10, L32, etc).
2) Definition of MDA (L118) and 3-NPA (L138) seemed lacking.
1) There are many typos and grammatical errors (ie L10, L32, etc).
Author Response
Dear Reviewer-3,
Thanks for your comments concerning our review entitled “Polyamines in ovarian aging and disease”. Those comments are all valuable and helpful for revising and improving our review. We have studied all comments carefully and have made correction. Revised portion are highlighted in the review. The main corrections in the review and the responds to the comments are as following:
Yours sincerely,
Bo Kang
E-mail: bokang@sicau.edu.cn
Response to Reviewer-3,
Comments 1: In contrast, the role of polyamines was mainly discussed on ovarian cancer based on gene regulation rather than oxidative stress.
Response 1: ODC is the rate-limiting enzyme for polyamine synthesis and a direct transcriptional target of MYC. The majority of cancers exhibit a significant increase in MYC expression, which in turn promotes ODC expression to expedite polyamine biosynthesis [1-3]. Spermidine is known to be the only substrate for eIF5A activation. It has been shown that hyp-eIF5A can regulate MYC expression to treat cancer [4, 5]. There are few biomarkers that are used for better understanding how oxidative stress is involved in cancer pathophysiology [6]. There are no reports of polyamines, oxidative stress and ovarian cancer. Therefore, ovarian cancer is not discussed in our manuscript based on the fact that polyamines play an anti-oxidative stress role.
Comments 2: Role of polyamine on prostaglandin E2 seemed obscure.
Response 2: Prostaglandin E2 is an important pro-inflammatory factor. Spermidine inhibits inflammation by suppressing LPS-induced prostaglandin E2 release from mouse RAW264.7 macrophages [7].
Major concerns:
Comments 3: In Fig. 3, it seemed unclear the role of other antioxidants. It is important to described how polyamines can act in text, if this is important. Is Fig 3 essential for this manuscript?
Response 3: Figure 3 is necessary for this manuscript. We have added to the review that polyamines exert their antioxidant effects through other pathways.
Minor concerns:
Comments 4: There are many typos and grammatical errors (ie L10, L32, etc).
Response 4: Revised in line 10, 32.
Comments 5: Definition of MDA (L118) and 3-NPA (L138) seemed lacking.
Response 5: Revised in line 119, 151.
Comments on the Quality of English Language
Comments 6: There are many typos and grammatical errors (ie L10, L32, etc).
Response 6: Revised in line 39, 56, 57, 120, 137, 139, 164, 186, 187, 201, 205, 247, 296, 311 and 378. And we apologize for the poor language of our manuscript. We worked on the manuscript for a long time and the repeated addition and removal of sentences and sections obviously led to poor readability. We have now worked on both language and readability and have also involved native English speakers for language corrections. We really hope that the flow and language level have been substantially improved.
References:
- Bachmann, A. S.; Geerts, D., Polyamine synthesis as a target of MYC oncogenes. J Biol Chem 2018, 293, (48), 18757-18769.
- Casero, R. A., Jr.; Murray Stewart, T.; Pegg, A. E., Polyamine metabolism and cancer: treatments, challenges and opportunities. Nat Rev Cancer 2018, 18, (11), 681-695.
- Bello-Fernandez, C.; Packham, G.; Cleveland, J. L., The ornithine decarboxylase gene is a transcriptional target of c-Myc. Proc Natl Acad Sci U S A 1993, 90, (16), 7804-8.
- Coni, S.; Serrao, S. M.; Yurtsever, Z. N.; Di Magno, L.; Bordone, R.; Bertani, C.; Licursi, V.; Ianniello, Z.; Infante, P.; Moretti, M.; Petroni, M.; Guerrieri, F.; Fatica, A.; Macone, A.; De Smaele, E.; Di Marcotullio, L.; Giannini, G.; Maroder, M.; Agostinelli, E.; Canettieri, G., Blockade of EIF5A hypusination limits colorectal cancer growth by inhibiting MYC elongation. Cell Death Dis 2020, 11, (12), 1045.
- Coni, S.; Bordone, R.; Ivy, D. M.; Yurtsever, Z. N.; Di Magno, L.; D'Amico, R.; Cesaro, B.; Fatica, A.; Belardinilli, F.; Bufalieri, F.; Maroder, M.; De Smaele, E.; Di Marcotullio, L.; Giannini, G.; Agostinelli, E.; Canettieri, G., Combined inhibition of polyamine metabolism and eIF5A hypusination suppresses colorectal cancer growth through a converging effect on MYC translation. Cancer Lett 2023, 559, 216120.
- Jelic, M. D.; Mandic, A. D.; Maricic, S. M.; Srdjenovic, B. U., Oxidative stress and its role in cancer. J Cancer Res Ther 2021, 17, (1), 22-28.
- Jeong, J. W.; Cha, H. J.; Han, M. H.; Hwang, S. J.; Lee, D. S.; Yoo, J. S.; Choi, I. W.; Kim, S.; Kim, H. S.; Kim, G. Y.; Hong, S. H.; Park, C.; Lee, H. J.; Choi, Y. H., Spermidine Protects against Oxidative Stress in Inflammation Models Using Macrophages and Zebrafish. Biomolecules & therapeutics 2018, 26, (2), 146-156.

Round 2
Reviewer 3 Report
Dear the Editor
Raised concerns were properly addressed by this revision.